# Development of a Resource Optimization Platform for Cross-Regional Operation and Maintenance Service for Combine Harvesters

**Weipeng Zhang, Bo Zhao \*, Liming Zhou, Conghui Qiu, Jizhong Wang, Kang Niu, Hanlu Jiang and Yashuo Li**

State Key Laboratory of Soil Plant Machine System Technology, China Academy of Agricultural Mechanization Science Group Co., Ltd., Beijing 100083, China

\* Correspondence: caamsjds309@gmail.com

**Abstract:** In view of the centralized operation, high failure rate and large number of harvesters involved in the cross-regional operation of combine harvesters, which has led to a surge in maintenance service demand and a lack of effective maintenance service systems, in order to be able to quickly solve problems arising from failures during the process of cross-regional operation, an operation and maintenance (O&M) service platform for the cross-regional operation of combine harvesters was designed in this research on the basis of data resources, supported by the computing power of a big data platform and centered on an artificial intelligence algorithm. Meeting the demand for maintenance service during cross-regional operation, we built a system platform integrating service order management, maintenance service activity management, and maintenance service resource management, and a technical algorithm for operation and maintenance service resource allocation and service path optimization was developed in order to achieve service function modularization and intelligent monitoring, while early warning and display were realized using multi-dimensional platforms such as a PC, a control screen, and a mobile App. This platform was able to solve problems arising when harvesters break down, maintenance service can be carried out quickly when traditional resource information is blocked and the demand for the service is difficult to meet. The reduction in cost and the increased efficiency for agricultural machinery enterprises was also achieved, while the problem of ensuring continued service was systematically solved during the process of cross-regional operation. Finally, the performance of the software architecture and the effect of path optimization were verified. The results showed that the platform system developed using the three-layer C/S architecture offered more stable characteristics, and the path optimization in the platform system was better able to reduce the maintenance time and distance, thus making it possible to realize the dynamic on-demand configuration and scheduling management of cross-region job service resources.

**Keywords:** cross-regional operation of agricultural machinery; cross-regional maintenance service; resource scheduling optimization; operation and maintenance service platform

## 1. Introduction

Food security is an important context for work all over the world. The cross-regional operation of combine harvesters has led to a significant guarantee of food security in China. Due to China's geography, seasons and production environment, the harvest season in China is seasonal and time-phased [1]. Meanwhile, due to the agricultural environment in China, such as small plots of land being owned by farmers and the wide distribution of farmland [2], a combine harvester can not only satisfy the use needs of their owners, but also provide services for families without agricultural machinery, bringing economic income to the owners of agricultural machinery, and performing harvest tasks for other regions in China. Thus, agricultural machinery operation teams are constantly being formed, and agricultural machinery enterprises provide these operation teams with order information and guarantee the maintenance servicing of the agricultural machinery through

the management of resources and agricultural machinery [3,4]. Agriculture 4.0 marks a rapid increase in connection and the sensing of various information resources across regions, resources and systems [5]. The effective allocation and deployment of agricultural service resources is key to accelerating the rapid development of the post marketization of agricultural machinery. Traditional agricultural machinery maintenance services, mostly based on manual experience, adopt self-repair while finding other similar mechanical equipment maintenance shops nearby when encountering problems related to the failure of agricultural machinery. However, due to the lack of knowledge regarding agricultural machinery, it is difficult to solve problems related to the failure of agricultural machinery, while at the same time, this affects the efficiency and duration of agricultural machinery operation. The effective collaborative management of various elements in agricultural machinery production service scenarios can provide many innovative solutions for the modernization of the agricultural sector [6].

In European countries and the United States, most farms have their own maintenance workers and maintenance experience [7]. The development of the system platform in the agricultural scene can effectively reduce costs and reduce resource waste. Agricultural machinery has been brought into the sharing economy [8]. Considering the time window, operation and maintenance cost, and operation and maintenance distance, the collection and configuration management of agricultural machinery service resources can effectively be used to solve the service needs of small farmers in developing countries, such as the need for the maintenance of agricultural machinery. As an important part of the whole supply chain, the agricultural business field is expected to play an important role in improving agricultural activities. The intelligent collaborative management and control of farm management tasks [9] requires a more comprehensive activity monitoring system and information system. Research institutions and scientific groups have made continuous efforts to provide solutions and products to solve different agricultural problems by means of the Internet of Things [10], including by performing research on sensor networks, agricultural facility management and control systems, and farm production environments [11]. Interoperability between services and stakeholders is promoted through a novel information architecture [12]. Information communication technologies (ICT) such as farm management information systems (FMIS) have brought significant improvements in terms of productivity, providing management advantages through applications such as sensing, data mining, and analysis [13]. Innovative service processes can be used to improve the current integration of resource information, improve service response speed and time, and interact by accessing the same data set in the cloud. Agricultural machinery enterprises can quickly solve the maintenance service of agricultural machinery in cases of sudden failure [14] through the effective combination and systematic management of operation and maintenance service resources. Comprehensive operation and maintenance are further divided into preventive maintenance [15], predictive maintenance, and fault maintenance. Cloud platform systems have the ability to perform data analysis, possess large amounts of storage management space, and offer a unified management platform that is able to schedule tasks and events in real time, most of which are concentrated in the fields of logistics distribution, emergency rescue, and so on [16–18]. John et al. [19] defined agricultural digital services (DSAs) as the solution of agricultural challenges by using digital devices such as mobile phones, computers, satellites, and sensors. Some of the agricultural management platforms designed and developed by domestic scholars listed in Table 1 have greatly promoted the research and development of intelligent dispatching systems and the improvement of their functions. From the literature, it can be found that there is a lack of effective system platforms for the cross-regional operation service of combine harvesters in view of the typical large-scale migration of agricultural machinery, and it is urgent to solve this problem in order to achieve improved performance in terms of both the architecture and the actual effect of the system platform.

To solve the problems of loose cross-regional maintenance resources and the low use rate of combine harvesters, a cross-regional operation and maintenance system platform

for agricultural machinery maintenance services is developed in this study, and the system performance of the adopted architecture is compared in terms of software architecture. The path optimization effect was verified in the system, demonstrating that the path optimization performance of the developed platform was better than that of the traditional random path and time situation. In the end, the platform system is able to quickly solve the problem of agricultural machinery failure by forming a framework of database information, service resource management, resource allocation management and service application, in order to effectively coordinate maintenance service resources.

**Table 1.** Research and design related to agricultural machinery management platforms.

| Type of Agricultural Dispatch Platform | System Function | Source in Literature |
| --- | --- | --- |
| Farm management system platform | The adoption of the new agricultural system model created a market for services and applications available to farmers. | [20,21] |
| Agricultural machinery dispatching system platform | Job scheduling method can be carried out through the system platform. | [22–24] |
| Agricultural Internet of Things platform | Realized the networking management and control of each module in the new agricultural scene. | [25–27] |

## 2. Analysis of Demand for a Cross-Regional Operation and Maintenance Service Platform for Harvesters

### 2.1. Research Background

In China's summer harvest season, more than 650,000 combine harvesters are put into use throughout the whole country, and 250,000 are involved in cross-regional operations [28]. Depending on the seasonal change in China, there are fixed time requirements for harvesting and sowing. If the sowing and harvesting time is missed, the yield will be affected. Emergency harvest and seed service teams can be established in order to effectively guarantee the grain harvest [29]. Managing and controlling the optimization of operation and maintenance service resources in agricultural scenarios can resolve harvester failures quickly, helping to improve overall operation and maintenance efficiency. It can be found from the literature that in light of the typical large-scale migration of agricultural machinery operation, there is a lack of effective system platforms for cross-regional operation service of combine harvesters, and the performance of the architecture, as well as the actual effect of the system platform, need to be urgently addressed.

### 2.2. Analysis of Requirements

At present, in view of the formation of cross-regional operation service teams of combine harvesters, there is a lack of overall planning and systematization for support maintenance services. The sudden failure of a combine harvester can affect the harvesting efficiency of agricultural machinery operators and the harvest time of field crops, and there is a lack of unified workflow management and data flow management among the various maintenance resources. At the same time, due to the migration of harvester operation service teams and the lack of information resources with respect to service sites and maintenance stations in new locations, it is difficult to quickly solve fault problems and effectively obtain information regarding the availability of maintenance personnel, maintenance stations and spare parts. Due to the lack of maintenance experience among operators, they are only able to seek professional harvester maintenance personnel for the purpose of solving the problem. With the mastery of information regarding service providers possessed by agricultural machinery enterprises, as well as the growing service teams of combine harvesters, the increasing demand for information fusion of various service resources, the interoperability between various vertical systems is becoming increasingly necessary, and it is increasingly urgent that the system is able to communicate and

share data seamlessly, realize information sharing, and achieve information consistency. As shown in Figure 1, the sensor monitoring devices installed on combine harvesters produced by agricultural machinery enterprises and the existing CDMA/GPRS network technology can be used for the transmission of fault information, service resource information analysis, and the timely transmission of maintenance work orders, thus providing a convenient service platform for cross-regional operation service teams for combine harvesters, and effectively solving the problem of performing maintenance work for the cross-regional operation of combine harvesters in China every year.

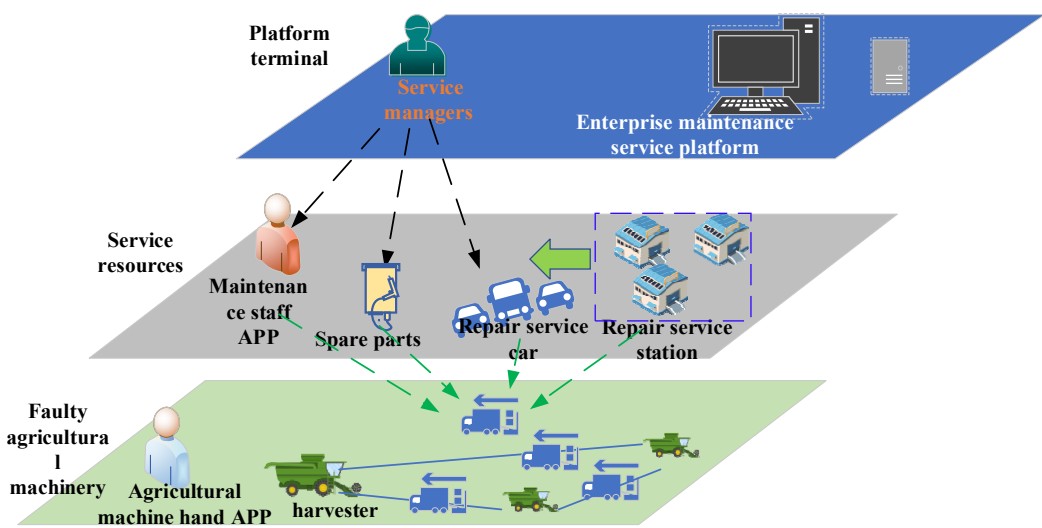

**Figure 1.** Schemes follow the same formatting.

## 3. System Design

### 3.1. Overall Design of the Platform

A management framework for operation and maintenance based on the cross-regional operations of combine harvesters is put forward in the form of the platform depicted in Figure 2. The developed system had four layers of architecture, namely, the support layer, the data layer, the functional layer, and the user layer. The support layer was mainly responsible for the collection of sensing data acquired from combine harvester, data transmission from the vehicle terminal of the combine harvester, and server storage and loading of the software, all of which are used to establish, store and combine data related to remote operation and maintenance, the operational information of key components, and data on the status of the combine harvester. By means of the support layers, the application logic for harvester repair services can be quickly created, assembled, deployed and managed. The data layer comprises a database for service resource operation and maintenance constituted on the basis of service branch data, maintenance personnel data, data related to spare parts centers and spare parts quantities, maintenance service vehicle data, and expert maintenance history data, all associated and structured as SQL database service resources. The permission settings of the O&M platform, the business logic relationships of each module, and the workflow of maintenance and management activities were also written using the SQL database. The functional layer mainly uses the intelligent optimization algorithm according to the maintenance service scenario and the fault maintenance task order to quickly complete maintenance work in consideration of factors such as maintenance path, cost, and time. The service interface converts the basic data in order to realize business functions, including real-time records of operation and maintenance management activities, visual analysis and early warning displays of faults in combine harvesters, traceable queries, statistical analysis, etc. In the user layer, the platform functions are integrated and classified for users, and finally integrated for three major users: platform management personnel, maintenance personnel, and agricultural machinery drivers.

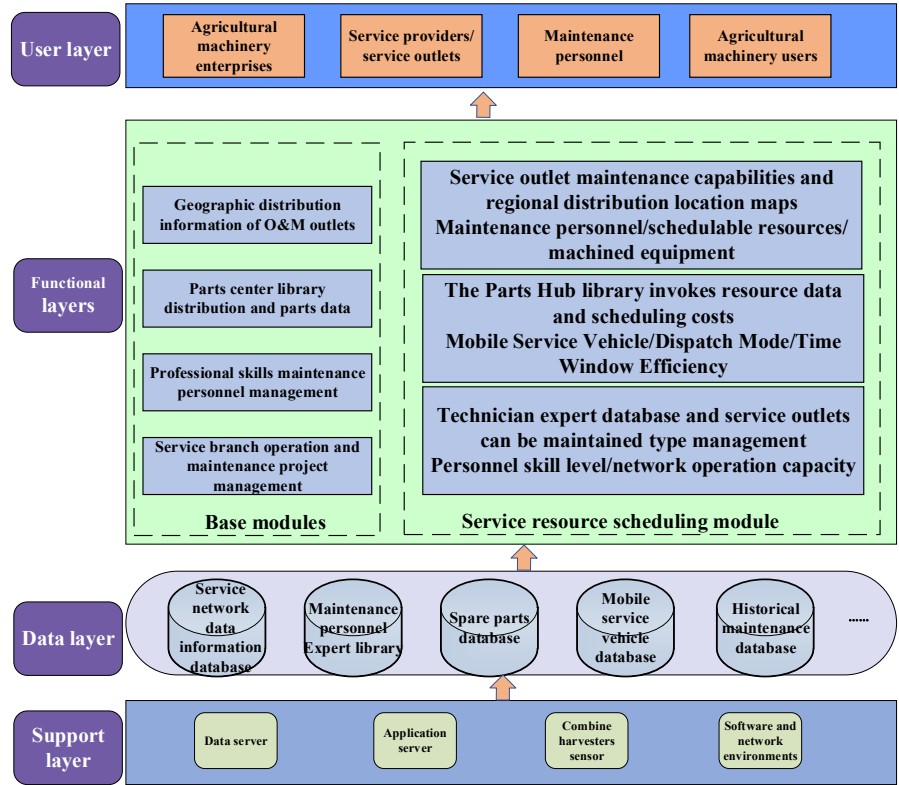

**Figure 2.** Overall design structure diagram of the platform.

### 3.2. Functional Design of the Platform

The system functions comprised four main parts: resource management and configuration module, order management module, operation and maintenance mode module and user management module. The functional architecture is shown in Figure 3.

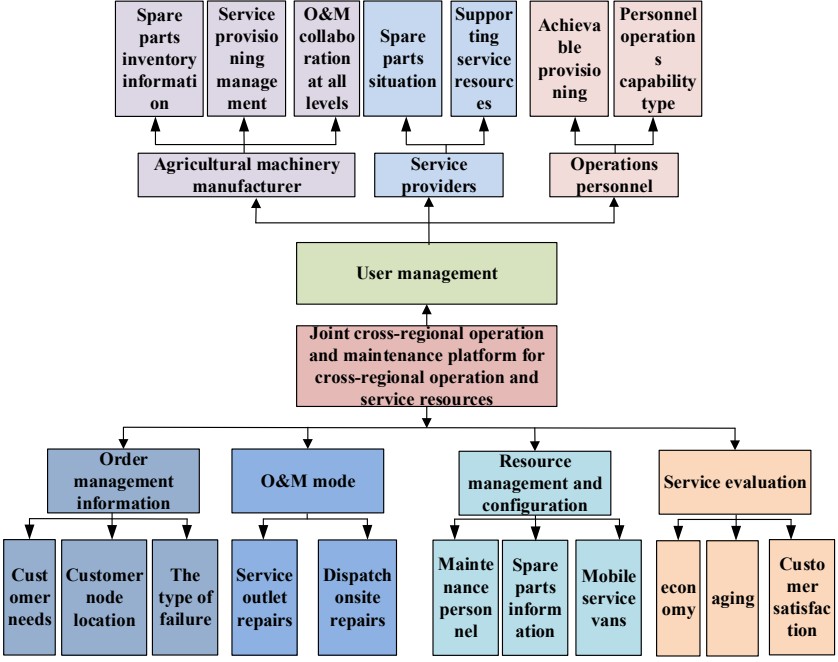

**Figure 3.** System functional structure.

Order management information mainly refers to sorting out customer information in order to create a maintenance request order after maintenance requests sent by the agricultural machinery operator have been received. The customer information is sorted out, including information such as the location, cause and type of the reported fault, and is then used to conveniently allocate maintenance resources to the harvester in order to provide maintenance.

There are two main operation and maintenance modes in maintenance scenarios: service network maintenance and scheduling maintenance. Service network maintenance sends a query to the agricultural machine operators, displays the maintenance resources available in the service area through the app, and provides information on the available maintenance network for agricultural machine operators, as well as the available spare parts available in the current area. Maintenance scheduling mainly provides regional maintenance network information, maintenance personnel information, and information on replacement spare resources with respect to the fault maintenance location reported by the agricultural machinery operators. Agricultural machinery operators are able to choose between active maintenance and prompt maintenance using the platform.

As the main core of the platform, the resource management and configuration module includes the information on the execution of service for the entire maintenance scenario, including information on maintenance personnel, service vehicles, spare parts, etc. The typical maintenance service mode is that in which maintenance personnel drive a maintenance service vehicle to the location of the failure with the spare parts corresponding to the required maintenance in accordance with the report describing the faulty agricultural machinery and order information received by the app. Multiple orders will generally be assigned to maintenance personnel. A dynamic optimization algorithm was used to optimize the order, maintenance route and time required for service provision, and service operation and maintenance work were carried out in consideration of the minimum maintenance cost.

User management is performed with reference to the information on personnel included in the operation and maintenance platform, including information regarding the quantity of spare parts produced by the first-level manufacturer of agricultural machinery, information regarding increases or decreases in the configuration of each service unit, and the details of collaborative operation and maintenance at all levels, including the spare parts available from the spare parts center library, the spare parts production library and the regional configuration service library.

Evaluation refers to customers' evaluation of the operation and maintenance service process, where the commencement of the maintenance task is signalled by order generation. On the basis of the fault resolution time and the maintenance cost involved in the service process, agricultural machine operators are able to evaluate the tasks through the app in order to provide feedback for the platform. The evaluation information serves to represent the level of the quality of the maintenance operation and the star rating given to the maintenance personnel and service provider.

### 3.3. Functional Architecture Design of the Platform

In the field of software development, application systems with C/S architectures possess strong logical transaction processing and data processing capabilities, strong interactivity, fast response, and are able to perform complex business processes. Systems with B/S architectures possess strong distribution and sharing, as well as convenient maintenance processes, and are able to realize cross-platform and cross-application data sharing. The software design of the cross-regional operation and maintenance platform for combine harvesters adopted a design mode using a three-tier C/S architecture, which is able to intelligently and comprehensively manage and monitor the long-distance O&M of the combine harvesters. The operation and maintenance platform makes decisions regarding the analysis and output of the data of each harvester module, and develops a complete historical data report (including operating parameters, curve charts, and alarm records). When a problem arises with an online combine harvester, the platform is able to quickly

send the fault information to the driver via the app, while at the same time pushing maintenance and service plans to the driver. The platform is able not only to collect real-time information using the harvester's monitoring module, it is also able to query and access data information by means of service platforms and mobile terminal apps for agricultural machinery operators and maintenance personnel, while at the same time, it is also able to push relevant technical parameters to the relevant personnel through text messages and telephones. The functional architecture of the platform for the cross-regional operation and maintenance service for combine harvesters is shown in Figure 4.

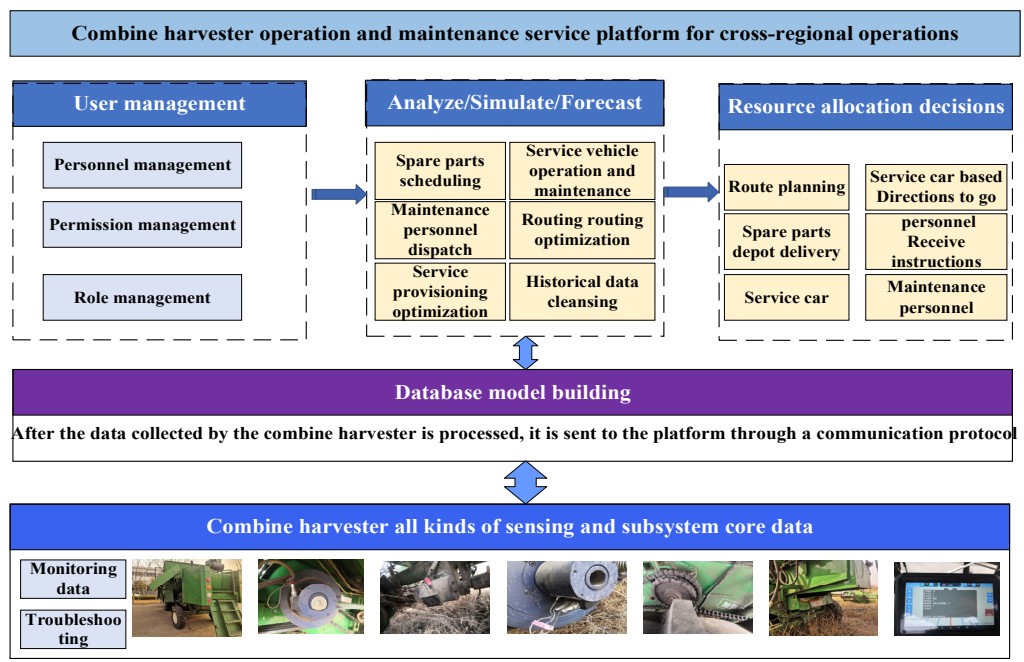

**Figure 4.** Functional architecture of the inter-regional operation and maintenance service O&M platform.

## 4. Platform Development of Key Technologies

### 4.1. Optimization Technology for the Resource Allocation of O&M Services

As shown in Figure 5, in view of the diversified service demands of combine harvesters under different operation and maintenance modalities, the density of agricultural machinery in regional operations was taken as the main basis in the current study. Considering factors such as production seasonality, workload, regional environment, operating conditions, and agricultural machinery status, operation and maintenance forecast information was adopted on the basis of historical information and big data analysis. Comprehensive resource information analysis and mining of available technology, such as with respect to cluster operation service requirements, multi-level service network distribution, spare parts reserves, and service vehicles and personnel, were studied, and the problem of the extensive allocation of service resources was solved, thereby improving the effectiveness of resource allocation.

### 4.2. O&M Service Path Optimization Analysis Technology

As shown in Figure 6, based on the analysis of information from multiple sources, such as fleet demand, network distribution, personnel, and spare parts reserves, and in order to determine the number and type of service orders, technologies for scheduling resources such as service outlets, spare parts, personnel, and mobile service vehicles in the regional environment were studied. Operation and maintenance service work requires the maximization of time efficiency, so it is crucial that the scheduling of maintenance services be performed in the shortest possible time. Depending on the reports and determination of service information obtained from the agricultural machine operator, information is

sent to the operation and maintenance service platform, and an optimized maintenance service order will then be sent to maintenance personnel. This allows the optimal service path to be provided to the maintenance personnel, and the function takes into account the distance between the fault demand point and the service station and the maintenance service vehicle, so that emergency relief materials can be transported to the destination in the shortest possible time and at the lowest cost.

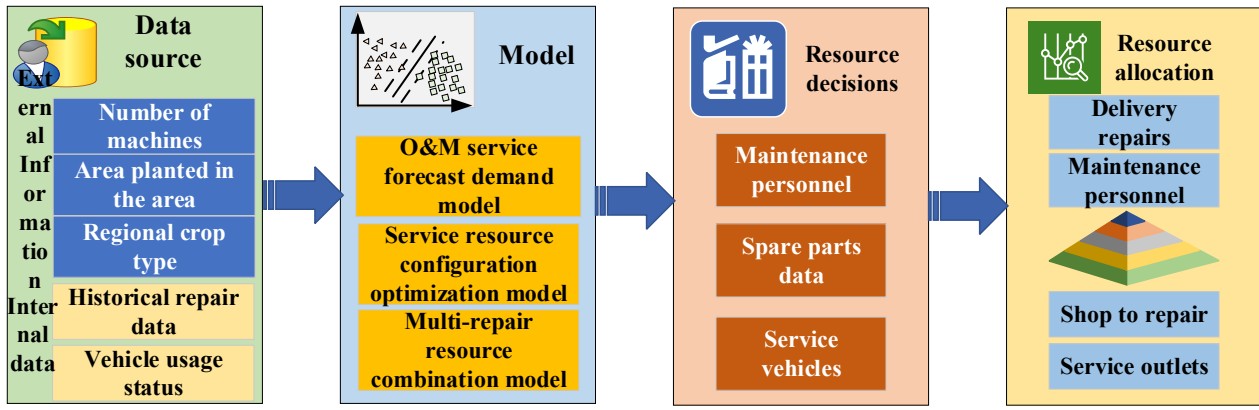

**Figure 5.** O&M service resource architecture diagram.

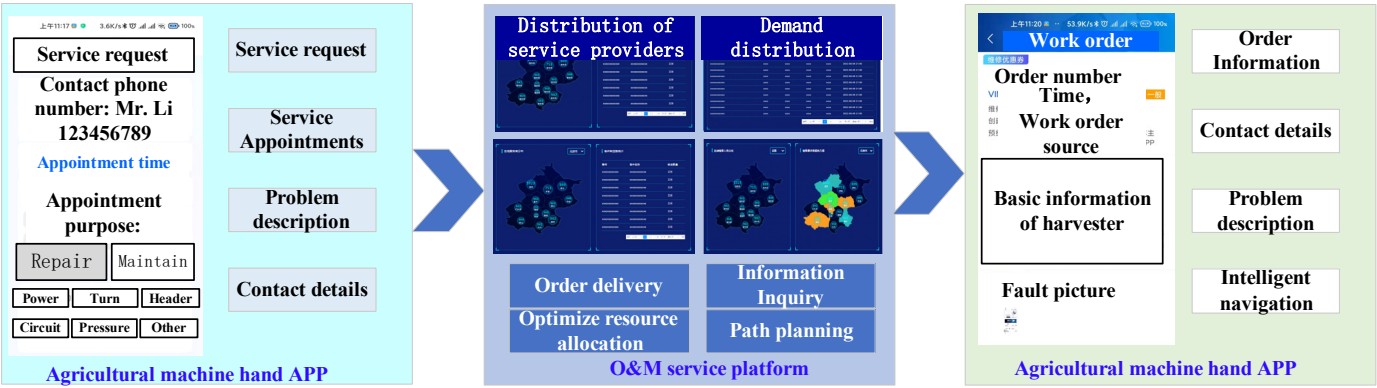

**Figure 6.** O&M service path optimization flowchart.

## 5. O&M Service Platform Implementation

The system was based on the visualization, networking, and development of an environment that integrated the information query and editing, while incorporating vehicle positioning and navigation, dynamic path scheduling for combine harvester service O&M, and involved the development of an intuitive, clear and friendly operation interface. The system adopted a three-layer Client/Server structure and an IIS deployment mode, making it possible to expand system function while ensuring security. The tools used to develop the system were Javascript, Visual C++, ASP scripting language, etc., using Oracle large database as a background, and the improved Dijkstra algorithm was adopted in order to optimize the distribution route, so as to achieve an optimal scheduling optimization strategy.

### 5.1. Development of Key Technologies for the Platform

As the implementation goal of the system, the maintenance service work order management of the operation and maintenance service platform included the name of users who had requested agricultural machinery maintenance, the source of the work order and the reported fault type (reported by the agricultural machinery operator/monitored by remote system), the status (to be repaired/being repaired/repaired), the service provider

receiving the order, the name of the maintenance worker, and the time at which the work order was created. The details of the work order can be viewed, including information on the location of the agricultural machinery. The work order is sent to the maintenance personnel app by the operation and maintenance service platform so that the information related to the maintenance work order can be consulted, and the instructions of the maintenance work order can be quickly carried out.

### 5.2. Maintenance Service Activity Management System

As shown in Figure 7, the maintenance service management system of the operation and maintenance service platform generates the optimal operation and maintenance service scheme dependeing on the situation of ongoing maintenance tasks, in consideration of constraints including the time window, operation and maintenance cost, and maintenance distance according to the location of the agricultural machinery failure, thus realizing the allocation of maintenance service vehicles and performing maintenance route planning. Meanwhile, upon the request of the operation and maintenance personnel performing the maintenance task, new maintenance work orders can be inserted through dynamic optimization scheduling, so as to maximize the satisfaction of the maintenance tasks.

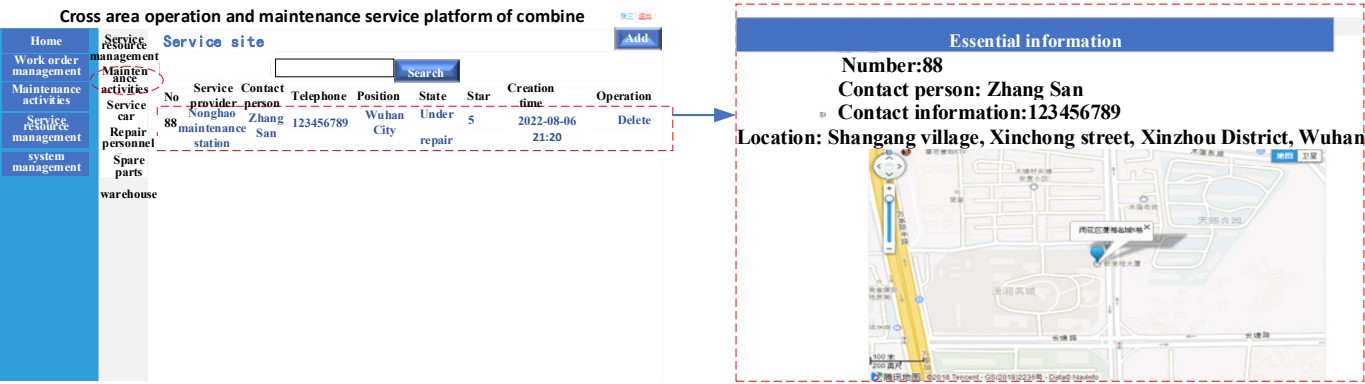

**Figure 7.** O&M service path optimization flowchart.

### 5.3. List of Repair Service Activities and Detailed Interface

The management of maintenance service resources for combine harvesters includes the management of maintenance service vehicles, the management of maintenance personnel, and the management of spare parts resources. As shown in Figure 8a, the on maintenance service vehicle module includes the vehicle's license, the service provider, the historical record of participation in maintenance, contact information, status, whether it can provide cross-regional service, service vehicle type, service distance and time. As shown in Figure 8b, the maintenance personnel module of the operation and maintenance service platform includes the service provider to which the maintenance personnel belong, the number of completed maintenance tasks, the historical maintenance failure type, the service area they are responsible for, and any evaluation indicators. As shown in Figure 8c, the spare parts management module provides a way to quickly handle faults during the maintenance process, and includes the name of the spare part, the type of spare part, the quantity in stock, and the service provider to which the spare part belongs.

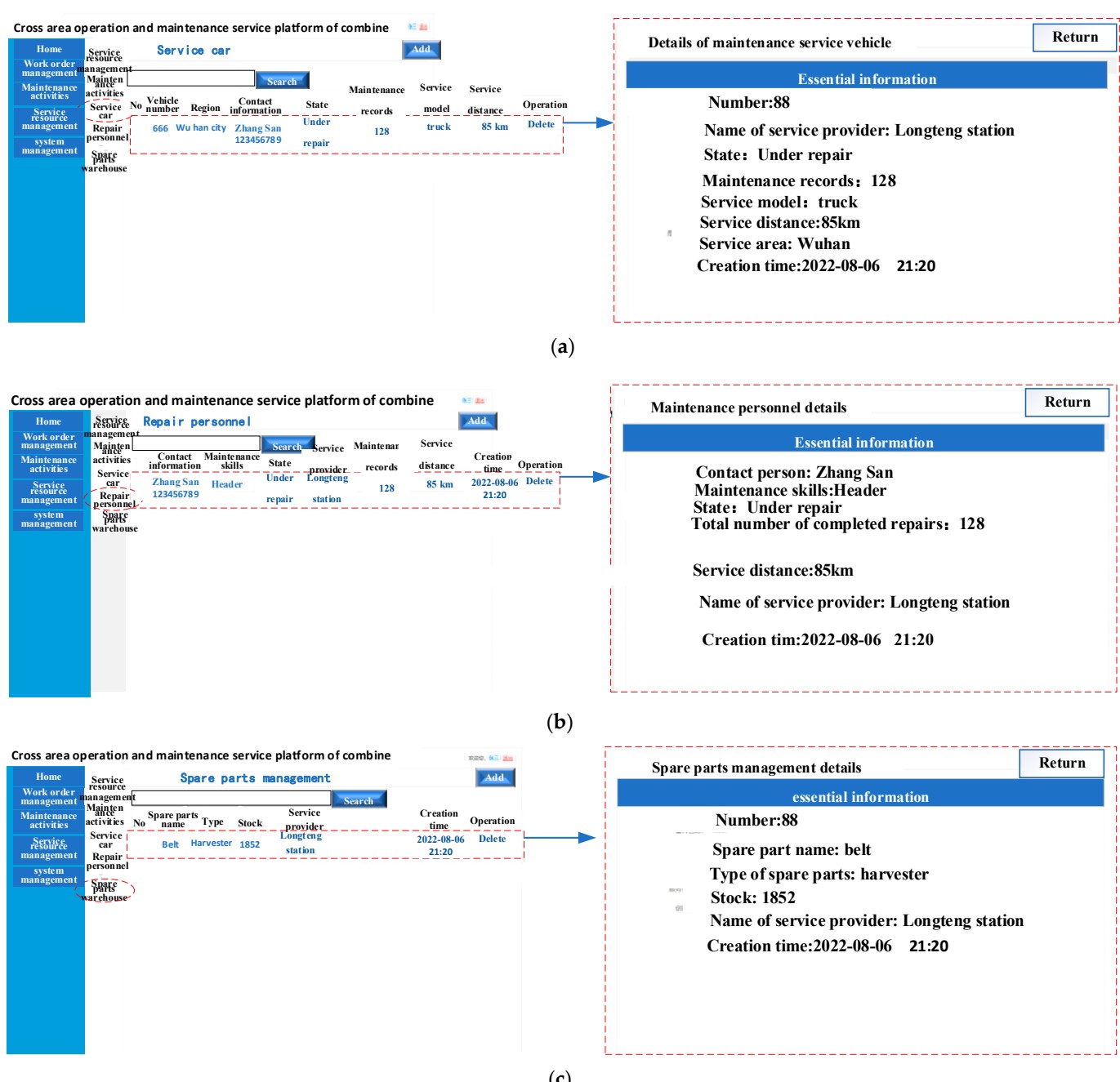

**Figure 8.** O&M service platform interface. (**a**) O&M platform service maintenance vehicle system interface. (**b**) O&M platform maintenance personnel system interface. (**c**) O&M platform spare parts information system interface.

## 6. Platform Application Validation

The research and application of the cross-regional operation and maintenance service platform for combine harvesters carried out in this paper served as the basis for the development of a support platform for the delivery of maintenance service during cross-regional operation of harvesters. On the basis of failures arising in combine harvesters, the fault diagnosis results generated by the on-board terminal of the combine harvester are uploaded to the operation and maintenance service platform. At the same time, the agricultural machine operator can also report information on agricultural machinery faults through the app. The service platform receives service requests generated by agricultural machinery operators, carries out fault analysis on the basis of an expert knowledge base,

and checks the operation and maintenance service resources in the area around the location of the fault. Through the optimization of the decision-making algorithm, a maintenance service request order is generated and send to the operation and maintenance personnel, and the operation and maintenance personnel carry out the maintenance service work according to the optimized service route generated by the platform. Figure 9 shows an implementation scenario of the service O&M platform. As can be seen from the figure, when a harvester fails, the optimal service scheme is distributed to the O&M personnel through the O&M service platform of the combine harvester, thus helping the O&M personnel to quickly reach the location of the harvester fault and carry out maintenance work.

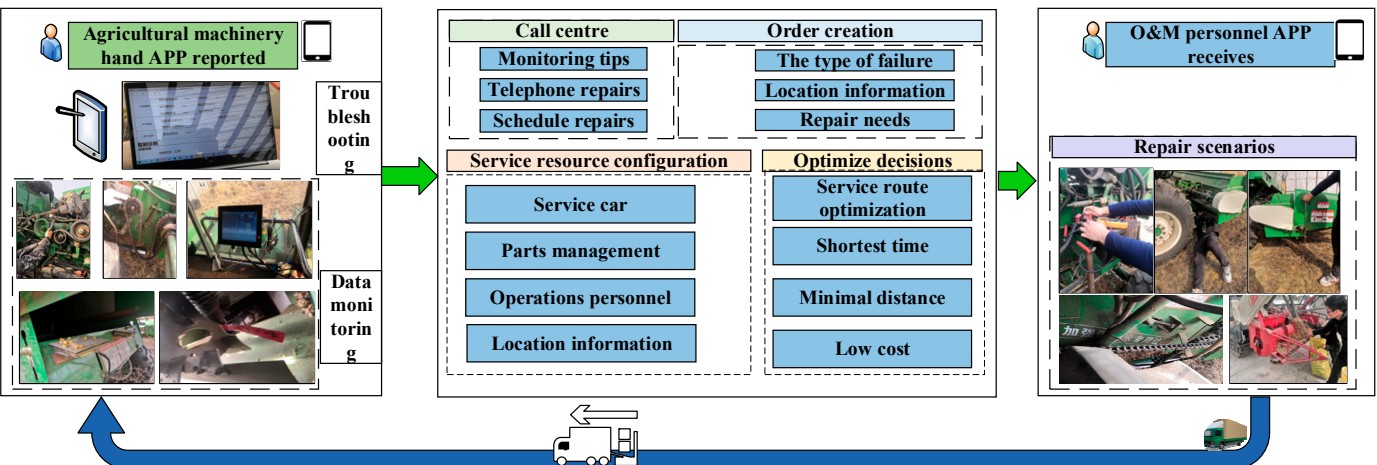

**Figure 9.** O&M service platform interface.

### 6.1. Platform Architecture Performance Validation

The research and application of the cross-regional operation and maintenance service platform of the combine harvester presented in this paper adopted a three-tier C/S architecture, and used Apache JMeter software to test the static and dynamic resource performance of the client and server, as shown in Figure 10. The traditional two-tier C/S architecture is a single server, with a local area network at its center. It is difficult to expand this architecture to a WAN in the case of large enterprises, or to the Internet. The combination and integration capabilities of software and hardware are limited by suppliers, making it difficult to manage a large number of clients. Therefore, a three-layer C/S architecture was developed. The three-tier C/S architecture divided the application function into three parts: the presentation layer, the functional layer, and the data layer. The three-layer division is clear and flexible, and is able to adapt to increased numbers of clients and changes in the processing load. C/S architecture, B/S architecture, two-tier C/S architecture, and three-tier C/S architecture were compared, and the results of the comparison are shown in Table 2.

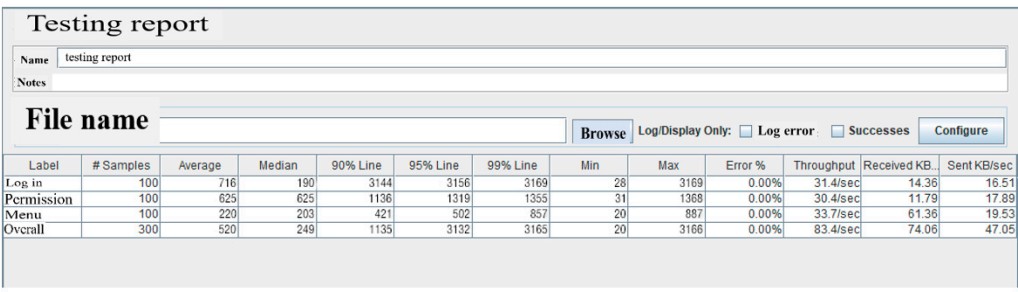

**Figure 10.** Platform architecture performance test.

**Table 2.** Comparison of four architectures.

| Architecture | Cross-Platform | Safety | Maintain | Responding Speed |
|---|---|---|---|---|
| B/S | good | poor | easy | slow |
| C/S | poor | good | difficulty | quick |
| Layer 2 C/S | good | poor | easy | faster |
| Layer 3 C/S | good | good | easy | faster |

As can be seen from Table 1, the three-tier C/S architecture offers a flexible hardware system composition, which improves program maintainability, is conducive to changing and maintaining application technical specifications, greatly increasing the security.

### 6.2. Verification of Path Optimization in the Platform

Taking the fault point of the wheat combine harvester as the research object, data related to the maintenance service vehicle and the faulty agricultural machinery were obtained from (Weichai Lovol Heavy Industry Co., Ltd., Weifang City, China). Firstly, the maintenance path of the wheat combine harvester was planned using the platform. The planning process combined the GPS positioning system with the maintenance process of the service vehicle. The position information was fed back in real time by the information system, and the maintenance operation tracking diagram is shown in Figure 11.

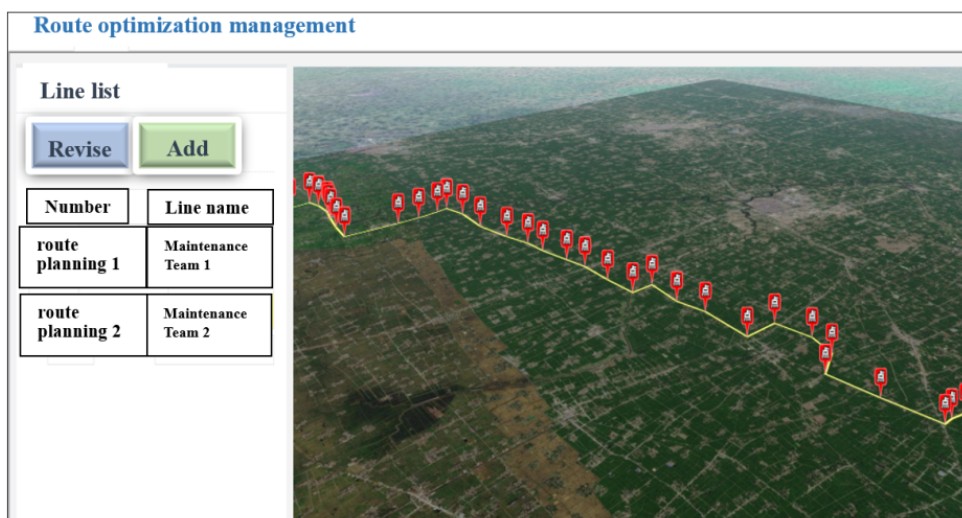

**Figure 11.** Route planning route map for the operation and maintenance service vehicle.

The Dijkstra algorithm and the A* algorithm were used to optimize the path of the map, and the results when using the two algorithms were simulated and compared. There are two main performance indicators for the purposes of comparison: search time and search accuracy. The map takes the location of each service vehicle and the faulty agricultural machinery as the node, and the time spent in each search is shown in Table 3.

**Table 3.** Comparison of search time between the Dijkstra algorithm and the A* algorithm.

| Node | Dijkstra's Algorithm | A* Algorithm |
|---|---|---|
| 32 | 4 s | 4 s |
| 40 | 7 s | 6 s |
| 63 | 16 s | 8 s |
| 77 | 24 s | 11 s |

As can be seen from Table 3, when the number of nodes is small, the search speed of the Dijkstra algorithm shows little difference with that of the A* algorithm, but when the

number of nodes is large, the Dijkstra algorithm is slower than the A* algorithm, and with increasing numbers of nodes, the difference in speed becomes greater.

The Dijkstra algorithm and A* algorithm are used for path planning. For a map with $N$ nodes, other than the starting node, the remaining nodes are all to be planned. The number of nodes to be planned is $N - 1$. Since each node to be planned corresponds to a shortest path, $N - 1$ shortest path should be planned for each map with $N$ nodes. The accuracy function is:

$$F = \frac{X}{N - 1} \tag{1}$$

where $X$ represents the number of shortest paths obtained by the algorithm, and $N$ represents the number of nodes.

The search accuracy of the two algorithms is compared in Table 4.

**Table 4.** Comparison of the Dijkstra algorithm and the A* algorithm with respect to search accuracy.

| Node | Dijkstra's Algorithm | A* Algorithm |
|---|---|---|
| 32 | 31 (100%) | 25 (80.65%) |
| 40 | 39 (100%) | 31 (79.49%) |
| 63 | 60 (96.77%) | 39 (62.90%) |
| 77 | 75 (98.68%) | 44 (57.89%) |

Table 4 shows that when the number of nodes is small, the accuracy of the A* algorithm is high, but it decreases with increasing numbers of nodes. However, the Dijkstra algorithm, no matter how the number of nodes changes, maintains an accuracy rate greater than 96%. Although the search time is long, the data are reliable. For the cross-regional operation and maintenance service platform for combine harvestesters, the Dijkstra algorithm is more reliable and suitable. After adopting the the operation path optimization strategy calculated by this algorithm, there was a reduction in the maintenance service operation path, a shortening of the time required, and a reduction in cost.

## 7. Conclusions

(1) The maintenance requirements for the cross-regional operation of combine havesters were taken as the research object in this paper, and a maintenance service operation and maintenance system platform for the cross-regional operation of agricultural machinery was developed. The system includes a maintenance service order management module, a service resource information management module, and a service resource activity management module. The processing and application of information at each stage of the process of the cross-regional operation of agricultural machinery was realized using the platform, and the interaction of information from multiple sources was achieved, and an association was realized between harvester vehicle-mounted information collection sensors, service provider operation and maintenance management, spare parts warehouses, personnel warehouses, and the maintenance knowledge base, effectively solving the situation whereby it is difficult to respond in real time with service resources, resulting in waste due to surges in the numbers of failures during the cross-regional operation of harvesters.

(2) To improve the resource service ability of combine harvesters and to ensure their effective application in cross-regional operation environments, a resource scheduling operation and maintenance platform for the cross-regional operation of combine harvesters was constructed. A three-layer C/S architecture was adopted in the system. The test results showed that the adopted structure had strong stability and high security. On the basis of a comparison of the path optimization effect, the effect of the platform integration optimization algorithm was tested, and the problem of shortages in service resource scheduling was solved, thus laying a foundation for the intelligent realization of cross-regional operation service scenarios in the future.

(3) Firstly, in view of the performance of the operation and maintenance platform with respect to agricultural machinery, only the carrying capacity of a single enterprise

was considered in this study. In a case where multiple agricultural machinery enterprises form an agricultural machinery alliance, the data resources of agricultural machinery will be shared, and the service network will be unified, and the stability and security of the system platform would also be worth considering. Secondly, depending on the quantity of available farmland, historical agricultural machinery demand and agricultural machinery failures in different regions, in the future, according to the historical operation and maintenance service situation, the maintenance needs of a region can be effectively predicted, and resources can be placed in order to reduce their waste. Finally, for the scheduling optimization of agricultural machinery service resources, only the path of service maintenance vehicles to the failure point of the agricultural machinery was considered in this study. When agricultural machinery presents regional maintenance requests, dynamic maintenance service stations and mobile maintenance service vehicles can be considered for solving the problem of operation and maintenance costs caused by distances that are too great.

**Author Contributions:** Conceptualization, W.Z. and B.Z.; methodology, W.Z. and H.J.; writing—original draft, W.Z. and Y.L.; funding acquisition, L.Z.; project administration, B.Z. and L.Z.; formal analysis, W.Z. and J.W.; investigation, K.N.; software platform, H.J. and C.Q.; data curation and software, visualization, W.Z. and Y.L. All authors have read and agreed to the published version of the manuscript.

**Funding:** The work was sponsored by the National Key R&D Program Project of China (2020YFB1709603).

**Institutional Review Board Statement:** Not applicable.

**Informed Consent Statement:** Not applicable.

**Data Availability Statement:** Not applicable.

**Conflicts of Interest:** The authors declare no conflict of interest.

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
