# Peer review of "Development of a Resource Optimization Platform for Cross-Regional Operation and Maintenance Service for Combine Harvesters"

_applsci, doi:10.3390/app12199873_

Round 1

Reviewer 1 Report

The issues raised by the Authors are topical and interesting for the reader. It can be a source of inspiration for both theorists and practitioners. It can be a starting point for further scientific research.

The text is large and clear. The Authors logically conduct a scientific argument.

The title indicates the subject matter discussed in the text.

The abstract is well written, comprehensive and sufficiently concise.

The keywords are appropriate.

The introduction is well written, setting the topic in context. The purpose of the text and its structure should be clearly indicated.

The Authors have performed a critical review of the literature (31 references). It is moderately extensive but relevant to the subject matter discussed - over 83% - these are publications issued in 2018 or later.

The text contains two self-citations:

1)      Zhang, W.; Zhao, B.; Zhou, L.; Wang, J.; Niu, K.; Wang, F.; Wang, R., Research on Comprehensive Operation and Maintenance Based on the Fault Diagnosis System of Combine Harvester. Agriculture 2022, 12, (6), 893.

2)       Wang, N.; Ren, S.; Liu, Y.; Yang, M.; Wang, J.; Huisingh, D., An active preventive maintenance approach of complex equipment based on a novel product-service system operation mode. Journal of Cleaner Production 2020, 277, 123365.

On the basis of the conducted analyzes, the Authors should identify research problems, formulate research hypotheses and clearly indicate the basic research gaps.

The methodological part is missing.

In the text, the Authors should refer to the research hypotheses/problems that they must pose in the earlier part of the text.

The conclusions are based on the results of the previous analyzes. The Authors should clearly indicate the shortcomings of research and perspectives for further research.

To sum up: the text is interesting, but requires some corrections and additions.

Author Response

1.The issues raised by the Authors are topical and interesting for the reader. It can be a source of inspiration for both theorists and practitioners. It can be a starting point for further scientific research.

The text is large and clear. The Authors logically conduct a scientific argument.

The title indicates the subject matter discussed in the text.

The abstract is well written, comprehensive and sufficiently concise.

The keywords are appropriate.

Response: Thanks for reviewing the manuscript. For the comments made by the experts, I and the co-authors express our heartfelt thanks. Your comments are valuable for the article. Thank you again for your time and help to the manuscript.

2.The introduction is well written, setting the topic in context. The purpose of the text and its structure should be clearly indicated.

Response:Thank you for the experts' careful review of the introduction of the manuscript. The questions you raised are very helpful to the improvement of the quality of the manuscript. We have carefully adjusted the contents of the introduction, and in view of the purpose and structure of the manuscript research that appeared in the introduction, thank you for your valuable suggestions on the manuscript. We have carefully revised the manuscript and highlighted the changes in the manuscript by text tracking. Thank you again for your help.

3.The Authors have performed a critical review of the literature (31 references). It is moderately extensive but relevant to the subject matter discussed - over 83% - these are publications issued in 2018 or later.

The text contains two self-citations:

1)      Zhang, W.; Zhao, B.; Zhou, L.; Wang, J.; Niu, K.; Wang, F.; Wang, R., Research on Comprehensive Operation and Maintenance Based on the Fault Diagnosis System of Combine Harvester. Agriculture 2022, 12, (6), 893.

2)       Wang, N.; Ren, S.; Liu, Y.; Yang, M.; Wang, J.; Huisingh, D., An active preventive maintenance approach of complex equipment based on a novel product-service system operation mode. Journal of Cleaner Production 2020, 277, 123365.

On the basis of the conducted analyzes, the Authors should identify research problems, formulate research hypotheses and clearly indicate the basic research gaps.

Response:Thanks to the opinions of the experts, we supplemented the research questions in the manuscript, and based on the research questions, we have made research goals, clarified our contributions through method assumptions and verifications, and finally verified the feasibility of our key technologies through method comparisons.

4.The methodological part is missing.

Response:We thank the experts for their suggestions. In order to improve the quality of the manuscript, we have supplemented the relevant research methods, research content and research objectives in the text according to the opinions of the experts, rearranged the abstracts and conclusions, and supplemented the methods and results.

5.In the text, the Authors should refer to the research hypotheses/problems that they must pose in the earlier part of the text.

Response:Thanks to the suggestions made by the experts. Through the guidance of the experts, based on some assumptions and problems mentioned in the first half of the manuscript, we have earnestly supplemented the relevant actual situation in the manuscript. According to the proposed research hypothesis, we added related data and comparisons at the end of the manuscript.

6.The conclusions are based on the results of the previous analyzes. The Authors should clearly indicate the shortcomings of research and perspectives for further research.

Response:We thank the experts for their suggestions for revision. According to the suggestions of the experts, we have supplemented the relevant content in the conclusion part of the article, and pointed out the shortcomings of the article and the future research prospects. The questions raised by the experts have a positive effect on the manuscript and improve the manuscript. scientific and reference value.

7.To sum up: the text is interesting, but requires some corrections and additions.

Response:We sincerely thank the reviewers for their suggestions on our manuscript, and thank the experts for pointing out the problems in our manuscript. We have carefully revised the relevant content and made revisions in the manuscript corresponding to the problems one by one. Thanks again to the experts for their help.

Reviewer 2 Report

  1. Abstract is well written. However, it needs to have the work reflected in the paper. For instance, it needs to have mention of proposed algorithm.  
  2. In introduction, Problem Motivation should be highlighted and summarize the key contributions.
  3. Mention the Key advantages of proposed methodology in the introduction section. 
  4. In the Literature Survey Verify the Key Contributions and highlight the proposed ones which are very much useful for your work.
  5. Abbreviation full form is to be given in the first occurrence and then use it directly later on. Verify this in the entire paper
  6. Figure 2 inside content not clear.
  7. Problem statement is to be elaborated and mention the key application areas of the work
  8. Mention the key parameters considered in the performance Evaluation
  9. Mention the importance of Various Parameters in the performance improvement measures
  10. Check the References section properly. Maintain the reference names in Synchronization order.

Author Response

  1. Abstract is well written. However, it needs to have the work reflected in the paper. For instance, it needs to have mention of proposed algorithm.

Response: Thanks to the questions raised by the experts, we have supplemented the proposed algorithm and the verification results in the abstract section of the manuscript in accordance with the experts' revision suggestions. At the same time, in order to improve the quality of the manuscript, we also supplemented the research methods and significance of the manuscript in the introduction.

  1. In introduction, Problem Motivation should be highlighted and summarize the key contributions.

Response: Thanks to the experts for their suggestions on the manuscript. In view of the incomplete introduction and the lack of research motivation and problems, we have added relevant content in the introduction, including the research problems and the innovations and contributions made by the manuscript.

3.Mention the Key advantages of proposed methodology in the introduction section.

Response: Thanks to the experts for their time and help in reviewing the manuscript.  We have carefully sorted out the problems in the introduction. We supplemented the proposed research methods in the introduction, and added the advantages and comparative analysis of the methods in the text. And based on these, in the last part of the introduction we have highlighted our innovations and contributions.

4.In the Literature Survey Verify the Key Contributions and highlight the proposed ones which are very much useful for your work.

Response: Thanks to the suggestions made by the experts, in the introduction, we have carefully revised the contribution of the literature based on the referenced literature. At the same time, based on the learning from these references, we have identified problems, and based on these problems, we have clarified our research goals. Finally, we highlighted our contribution and value to these issues in the manuscript, which made our manuscript more meaningful for further research.

5.Abbreviation full form is to be given in the first occurrence and then use it directly later on. Verify this in the entire paper.

Response: Thanks for the guidance and help given by the experts, which can effectively improve the quality of our manuscript. We re-reviewed the content of our entire manuscript according to the opinions of the experts. For the abbreviated parts in the text, we used the full name for the first time, such as the O&M in the text. We have carefully revised the text to address these issues, and sincerely thank the experts for their help.

6.Figure 2 inside content not clear.

Response: Thanks for the problems pointed out by the experts. For the unclear content in Figure 2, we have carefully revised it, and re-reviewed the pictures in the text to avoid the same problems. Thanks again for the guidance provided by the experts.

7.Problem statement is to be elaborated and mention the key application areas of the work

Response: We thank the experts for their suggestions. We apologize for the existence of such problems in the manuscript. According to the opinions of the experts, we have supplemented the manuscript with detailed explanations for the problems. At the same time, we have re-added the introduction, the text and the conclusion. The focus of our work is explained, the purpose of our research is explained in detail, and the research questions and research objectives are supplemented in detail.

8.Mention the key parameters considered in the performance Evaluation

Response: We thank the experts for their questions about our manuscript, and express our sincere gratitude to the experts for their careful reading of our article. These are very helpful to the improvement of the quality of our manuscript. According to the instructions of the experts, we have increased the parameters in the performance evaluation. At the same time, we used text traces to express these parameters in the manuscript. For example, the key parameters mentioned are: the length of the path, the length of time and the cost. We re-supplement these key parameters in the text so that our manuscript can provide reference for researchers.

9.Mention the importance of Various Parameters in the performance improvement measures

Response: Thanks to the experts for their suggestions and help, and we have made serious revisions in the text according to the opinions of experts, and highlighted them. We have compared and verified the developed platforms and supplemented the verification content. At the same time, in the conclusion, we also described the effects of improved performance.

  1. Check the References section properly. Maintain the reference names in Synchronization order.

Response: Thanks to the reminders and suggestions given by the experts. We have carefully sorted out the reference order in the introduction part of the manuscript, and corresponded to the reference part to ensure that the reference order can be synchronized in the introduction.

Reviewer 3 Report

          1. Kindly reduce abstract sections and add some numerical comparison. 2. In the end of introduction section add research objective. 3. Kindly add literature survey with comparison table. 4. Proposed architecture is good. 5. If possible add some graphical comparison in the result section. 6. proof read required 7. Conclusions need minor updation

Author Response

Reviewer 3

  1. Kindly reduce abstract sections and add some numerical comparison.

Response: We thank the experts for their suggestions and help. For the problems that only exist in some parts, in the manuscript, we have deleted the description of the abstract part, and supplemented the experimental verification content in the manuscript, and increased the experimental comparison value comparison. Based on this, we have added a numerical comparison in the summary section as you suggested.

  1. In the end of introduction section add research objective.

Response:Thanks for the suggestions given by the experts. For the problem of incompleteness in the introduction, we added the research objectives and research significance at the end of the introduction, carefully reviewed the introduction, and revised the problems in the references. We highlighted our research contributions by comparing the research of scholars, and at the same time, according to the opinions of experts, we added necessary content to make our manuscript more scientifically meaningful and reference value.

  1. Kindly add literature survey with comparison table.

Response:Thanks to the experts for their suggestions and guidance. We carefully reorganized the survey for the references in the introduction, and presented them in the manuscript in the form of a table, and supplemented and adjusted the reference survey and comparison table according to the experts' suggestions.

  1. Proposed architecture is good.

Response:Thanks to the experts for their suggestions and help. We have carefully sorted out the maintenance demands and other issues arising from the centralized operation of agricultural machinery by investigating enterprises and agricultural machinery users according to the situation of agricultural maintenance scenarios. The operation and maintenance platform was considered to realize the unification of resource data, the flow management, reduce unnecessary waste of resources, and effectively and quickly solve the problem of fault demand.

  1. If possible add some graphical comparison in the result section.

Response:Thanks to the experts for reviewing our manuscript and giving valuable suggestions. In view of the lack of data validation and experimental content in the text, we have re-added relevant content in the text according to the guidance of experts to highlight the research significance of our manuscript, and thanks again to the experts who provided valuable guidance for the revision of our paper.

  1. proof read required

Response: Thanks for the guidance and suggestions put forward by the experts, we have carefully proofread the language, introduction references and pictures in the article, and adopted the method of tracking handwriting in the article, hoping to get the satisfactory recognition of the experts.

  1. Conclusions need minor updation.

Response: Thanks to the experts for their valuable suggestions. Through our revision of the full text of the manuscript, we have also adjusted the conclusions of the manuscript, including the lack of innovation in the conclusions and the significance of the research. We have carefully adjusted the manuscript In the conclusion section, and re-reviewed it.  Thanks again to the experts for their valuable suggestions and help.

Reviewer 4 Report

I congratulate the authors for approaching this interesting field of study and its challenges. The agricultural industry is one of the most important in many countries, such as the emerging ones mentioned by the authors.

I highlight some points that can be improved in the manuscript to justify publication in a journal.

- The abstract could be improved by specifically highlighting the method and the main results of the research.

- The objective of the study needs to be more clearly defined in the introduction section. A single sentence must contain the purpose of the study clearly and precisely.

- The authors say “In developed countries such as Europe and America”. Country or continent? Please specify further.

- The authors need to do a more robust literature review on the field of study. The current review is weak and incipient. Current and important references in the field were not cited.

- It was unclear whether similar platforms already exist, whether in the literature or not.

- Can the tool be used in any country, for any agricultural crop? This needs to be evidenced in the manuscript.

- Some figures need to be made available in better quality

- The current literature is full of tools associated with the concepts of digital agriculture, precision agriculture and agriculture, 4.0. There are many management tools being developed every day to assist farms in their numerous challenges. Although the authors highlight the benefits of their tool, for me it is not clear if it has a differential that cannot be found in the literature or in startups around the world.

- Although the authors present the case well, I believe that a section on empirical and theoretical implications would benefit readers of the manuscript/journal. In this section it needs to be evident how the tool helps practitioners and organizations and how it contributes to the theory of the field of study.

- Finally, the Conclusion section is weak and needs improvement. In the conclusions, the authors need to present the limitations of the research developed and present suggestions for future studies in the field of study.

Author Response

  1. I congratulate the authors for approaching this interesting field of study and its challenges. The agricultural industry is one of the most important in many countries, such as the emerging ones mentioned by the authors. I highlight some points that can be improved in the manuscript to justify publication in a journal.

Response:Thanks to the experts for their suggestions and help on our manuscript. We thank the reviewers for their time and energy for our manuscript. We have carefully corrected the questions and suggestions raised by the experts, and have re-edited the shortcomings in the manuscript. Thanks again to the experts for their Acknowledge and help.

  1. The abstract could be improved by specifically highlighting the method and the main results of the research.

Response:We thank the experts for their suggestions and help. The abstract is very helpful to the readers. Based on the opinions of experts, we have seriously improved the expression quality of the abstract, and reorganized the focus and logic of the abstract.

  1. The objective of the study needs to be more clearly defined in the introduction section. A single sentence must contain the purpose of the study clearly and precisely.

Response: Thanks to the suggestions made by the experts, we re-introduced and adjusted the content format to clarify the research content and research goals of our manuscript. We sorted out the introduction and references of the manuscript, clarified the research focus and goals of the manuscript, and thanked the experts for their comments.

   4.The authors say “In developed countries such as Europe and America”. Country or continent? Please specify further.

Response: Thanks to the problems pointed out by the experts. We have carefully reviewed the content of the manuscript, and through literature reading, we have clarified our expressed views, which referred to "Country", and we have also corrected this issue in the manuscript.

  1. The authors need to do a more robust literature review on the field of study. The current review is weak and incipient. Current and important references in the field were not cited.

Response: We thank the experts for their time and suggestions for our manuscript. For the introduction and reference content of the manuscript, we have reorganized the key points of the references and introduction sections, adjusted the relevant content in the manuscript, and adjusted important references. Thanks for the expert’s advice.

  1. It was unclear whether similar platforms already exist, whether in the literature or not.

Response:Thanks for the experts' review of our manuscripts and their doubts. When we wrote the manuscripts again, we carefully checked the relevant literature and clarified the development of our platform system. Although there is relevant platform system development, it is only limited to the agricultural machinery's own platform, and there is no research on some problems arising from agricultural machinery's cross regional operation. In our platform system development, more full cycle operation and maintenance functions are highlighted, including cross region path, cross region cost and cross region time.

  1. Can the tool be used in any country, for any agricultural crop? This needs to be evidenced in the manuscript.

Response:Thank the experts for thinking about our manuscript. We carefully analyzed the content and objectives of our research. The platform system we developed is rare because of the trans regional characteristics of China's agricultural machinery. At present, it was proposed in response to the problems such as the intensification and diversification of maintenance needs in the trans regional operation of China's agricultural machinery, and our object is the maintenance of agricultural machinery. The suggestions and help from experts are very helpful to us. For the wrong crops and models, we have adjusted the input and output of the platform.

8.Some figures need to be made available in better quality.

Response:Thank the experts for their suggestions on our manuscript. In view of the problem of picture quality in the text, we corrected the problem in this part, carefully revised the picture quality in the manuscript, and updated the relevant picture quality, so that readers and researchers can refer to the content of our manuscript more conveniently.

9.The current literature is full of tools associated with the concepts of digital agriculture, precision agriculture and agriculture, 4.0. There are many management tools being developed every day to assist farms in their numerous challenges. Although the authors highlight the benefits of their tool, for me it is not clear if it has a differential that cannot be found in the literature or in startups around the world.

Response:The expert's opinion is very correct. Thank you for your help to our manuscript. Our tool is a platform for trans regional agricultural machinery maintenance services based on China's summer harvest season. It is conducive to the support of a large number of agricultural machinery service teams. Thank you for your suggestions.

  1. Although the authors present the case well, I believe that a section on empirical and theoretical implications would benefit readers of the manuscript/journal. In this section it needs to be evident how the tool helps practitioners and organizations and how it contributes to the theory of the field of study.

Response:Thanks to the experts for their suggestions and help for our manuscript, we carefully revised the thinking put forward by the experts. We supplemented the role of our development tools and help for agricultural scenes and agricultural practitioners in the manuscript. We supplemented the relevant research theories according to the research field and content.

11.Finally, the Conclusion section is weak and needs improvement. In the conclusions, the authors need to present the limitations of the research developed and present suggestions for future studies in the field of study.

Response:Thank the experts for their suggestions, which can help to improve the quality of our manuscript. In response to the experts' suggestions, we adjusted the content of the conclusion in the manuscript, and pointed out the problems in our research and analyzed them. We also provided future research directions and executable research tasks for our research, providing reference for researchers.

Round 2

Reviewer 1 Report

I accept additions to the text. In my opinion, the text is suitable for publication.